

# *NRF2* polymorphism and susceptibility to ischemic stroke in a Chinese population

Pengyu Wang[1,2,*], Junxiu Lu[1,3], Min Wang[1], Guangming Wang[1,4] and Huaqiu Chen[5,*]

[1] School of Clinical Medicine, Dali University, Dali, Yunnan Province, China
[2] Clinical Laboratory, Baoding No.1 Central Hospital, Baoding, Hebei Province, China
[3] The People's Hospital of Junan, Linyi, Shandong Province, China
[4] Center of Genetic Testing, The First Affiliated Hospital of Dali University, Dali, Yunnan Province, China
[5] Clinical Laboratory, Xichang People's Hospital, Xichang, Sichuan Province, China
[*] These authors contributed equally to this work.

Corresponding author
Guangming Wang,
wgm1991@dali.edu.cn

## ABSTRACT

**Background**. Ischemic stroke (IS) is a major health concern in the Chinese population. Previous studies have highlighted the role of NRF2 in IS. This study investigates the association between NRF2 polymorphisms and IS susceptibility in a Chinese population.

**Methods**. This retrospective study included Chinese patients diagnosed with IS based on clinical symptoms, neurological examinations, and brain imaging findings from computed tomography or magnetic resonance imaging. Age- and sex-matched unrelated individuals with no family history of stroke, tumors, or genetic diseases served as controls. Peripheral blood samples were collected to genotype seven single-nucleotide polymorphisms (SNPs) in NRF2 (rs13005431, rs4893819, rs6721961, rs35652124, rs6726395, rs2364723, rs2706110) using the SNaPshot method. Binary logistic regression analysis was used to assess associations between these SNPs and IS risk. NRF2 and reactive oxygen species (ROS) levels in peripheral blood were measured. The relationship between rs35652124 and NRF2 expression was evaluated using expression quantitative trait locus (eQTL) analysis.

**Results**. All seven NRF2 SNPs conformed to Hardy–Weinberg equilibrium. Six SNPs (rs13005431, rs4893819, rs6721961, rs6726395, rs2364723, and rs2706110) showed no significant differences in distribution between the case and control groups ($p > 0.05$). However, the TC genotype of rs35652124 in the co-dominant model was significantly associated with increased IS risk. The distribution of this genotype aligned with trends observed in East Asia and the Chinese Han population but varied across other global populations. The CCTTGGC haplotype was the most common in both groups. Stratified analysis of rs35652124 showed no association with confounding factors such as age, sex, hypertension, diabetes, or lipid levels. NRF2 and ROS levels were higher in IS patients than in controls, but did not differ by rs35652124 genotype. Concurrently, eQTL analysis indicated that rs35652124 did not affect NRF2 expression in peripheral blood.

**Conclusion**. The NRF2 rs35652124 polymorphism is associated with IS susceptibility, suggesting it may be a potential genetic risk factor for IS.

## INTRODUCTION

Cerebral stroke is a cerebrovascular disease influenced by multiple factors. The blockage or rupture of blood vessels in the brain can cause a localized interruption of blood flow, triggering a series of pathological damage processes. In China, cerebral stroke is the leading cause of death and disability among adults, with the highest incidence rate worldwide (*Wang et al., 2022b*). Among the different types of strokes, ischemic stroke (IS), which results from intracranial vascular occlusion, is the most prevalent. The incidence of IS in China has increased from 117 cases per 100,000 individuals in 2005 to 145 cases per 100,000 individuals in 2019, with a prevalence rate of 1,700 cases per 100,000 individuals in the same year (*Wang et al., 2022a*). In Yunnan Province, the incidence of IS rose from 220 cases per 100,000 people in 1990 to 674 cases per 100,000 people in 2017. During the same period, the disability-adjusted life-years attributed to IS increased from 770.7 to 815.4 per 100,000 individuals (*Liu et al., 2023*). These trends indicate a growing disease burden in both Yunnan and China as a whole.

Several modifiable risk factors, including hypertension, smoking, diabetes, and hyperlipidemia, contribute to IS. However, genetic factors are also recognized as significant contributors to disease susceptibility. Single-nucleotide polymorphisms (SNPs) in the genome influence an individual's inherent risk of developing IS (*Boehme, Esenwa & Elkind, 2017*). With advancements in genomic technologies, the role of genetic factors in IS onset and progression has gained increasing attention. Multiple studies have reported associations between genetic polymorphisms and IS susceptibility (*Almeida, 2013*).

Nuclear factor erythroid 2-related factor 2 (*NRF2*) is a key regulatory molecule that protects cells against oxidative stress. It plays a vital role in maintaining redox homeostasis and overall brain function. In response to oxidative stress, *NRF2* becomes activated and regulates over 250 downstream target genes involved in oxidative homeostasis. These include heme oxygenase 1, NADPH quinone oxidoreductase 1, and catalase (*Xu et al., 2020*). A previous report suggests that excessive reactive oxygen species (ROS) in IS leads to NRF2 upregulation, and the activation of its downstream target genes helps counteract oxidative damage, underscoring the neuroprotective role of *NRF2* in IS (*Farina et al., 2021*). Certain SNPs in *NRF2* have been associated with disease susceptibility. For example, the 617 C > A SNP has been linked to an increased risk of oxidant-induced acute lung injury (*Marzec et al., 2007*). Additionally, patients with cholangiocarcinoma who carry the *NRF2* rs6726395 GG genotype have a longer median survival time (344 ± 138 days, 95% confidence interval (CI) [73–615] days) compared to those with the AA/AG genotype (172 ± 37 days, 95% CI [100–244] days) (*Khunluck et al., 2014*). Recent studies have also demonstrated associations between *NRF2* SNPs and susceptibility to alcoholic liver disease (ALD), chronic kidney disease (CKD), and cardiovascular disease (*Adam et al.,*

*2017*; *Jerotic et al., 2019*; *Nunes dos Santos et al., 2019*). However, the relationship between *NRF2* polymorphism and IS risk in the Chinese population remains unexplored.

In this study, we investigated the correlation between *NRF2* SNPs and IS susceptibility in a Chinese population from Yunnan. We further analyzed the impact of these SNPs on NRF2 expression and examined their effects on ROS levels in the peripheral blood of patients with IS.

## MATERIALS & METHODS

### Study design

This study employed a retrospective design and was approved by the Medical Ethics Committee of The First Affiliated Hospital of Dali University (Approval no. DFY20220415001). All participants were informed of the study's purpose and provided verbal consent.

### Study participants

Using the Power/Sample Size calculation tool (http://www.stat.ubc.ca/~rollin/stats/ssize/caco.html), the statistical power was set at 80% with a two-sided alpha of 0.05. Based on the minor allele frequencies (MAF) and relative risk values of the loci in the Asian population, the minimum required sample size for both the case and control group was determined to be 135 participants each. Between March 2022 and October 2022, we recruited 151 patients diagnosed with IS who met the study criteria, as well as 141 healthy individuals who underwent physical examination at the First Affiliated Hospital of Dali University during the same period. All participants were unrelated. The inclusion criteria for the IS group consisted of patients diagnosed by a neurologist based on clinical symptoms, neurological function examination, and brain imaging (computed tomography or magnetic resonance imaging) in accordance with the 2018 Chinese Guidelines for the Diagnosis and Treatment of Acute Ischemic Stroke (*Chinese Society of Neurology Chinese Stroke Society, 2018*). The control group consisted of age- and gender-matched individuals (age and sex distributions were analyzed based on preliminary experimental data) with no history of cardiovascular or cerebrovascular diseases, autoimmune disorders, malignant tumors, immunologic diseases, neurological deficits, or severe hepatic or renal dysfunction.

### SNP selection and genotyping

Reference information for *NRF2* SNPs was obtained from dbSNP (https://www.ncbi.nlm.nih.gov/snp/). The selection criteria for SNPs included the following: presence in *Homo sapiens*, MAF > 5%, prior publication in scientific literature, inclusion in the 1,000 Genomes Project. Based on these criteria, seven SNPs were selected for analysis: rs13005431, rs4893819, rs6721961, rs35652124, rs6726395, rs2364723, and rs2706110. Peripheral blood samples (five mL) were collected from each participant in ethylenediaminetetraacetic acid-containing test tubes and stored at −80 °C. Genomic DNA was extracted using a blood genomic DNA extraction kit (Tiangen Biochemical Technology Co. Ltd., Beijing, China). DNA sample quality and concentration were determined using a NanoPhotometer N60 Touch microspectrophotometer (IMPLEN, Munich,

Germany). Genotyping of the seven SNPs was performed using the SNaPshot method on an ABI 3730XL sequencer (Applied Biosystems, Waltham, MA, USA). Experimental data management and analysis were performed using Peak Scanner software (v 1.0; https://www.thermofisher.com/order/catalog/product/4381867).

## Primer design and synthesis

Primers for the seven SNPs in *NRF2*, including polymerase chain reaction primers and unique base extension primers, were designed using Oligo (version 7.37; https://www.oligo.net/). The primers were synthesized by General Biology Co., Ltd. (Anhui, China). The sequence of the primers used are listed in Tables 1 and 2.

## Enzyme-linked immunosorbent assay (ELISA)

The levels of NRF2 and ROS in human peripheral blood were measured using ELISA with the double-antibody sandwich method. The assay kits were obtained from Shanghai Keshun Biotechnology Co., Ltd. (Shanghai, China) and used according to the manufacturer's protocol. Absorbance was measured at 450 nm using an Infinite 200 microplate reader (Tecan, Männedorf, Switzerland). The absorbance values were converted into corresponding concentrations based on the standard curve. The intensity of the developed color was directly proportional to the concentration of the target analyte.

## Bioinformatic analysis

The tissue-specific expression of NRF2 and expression quantitative trait locus (eQTL) analysis were performed using GTEx (http://www.gtexportal.org/). The global distribution characteristics of the selected SNPs were analyzed using Ensembl (http://asia.ensembl.org/index.html). Haplotype analysis of the seven SNPs was conducted using online haplotyping analysis software (http://analysis.bio-x.cn/myAnalysis.php) (*Shi & He, 2005*).

## Statistical analyses

All statistical analyses were performed using SPSS software (version 22.0; IBM SPSS Inc., Armonk, NY, USA). Normality testing was conducted for all quantitative data. If the data followed a normal distribution with equal variance, Student's *t*-test was applied; otherwise, the Mann–Whitney U rank-sum test was used. Ordinal data were analyzed using the Pearson $\chi^2$ test. The Hardy–Weinberg equilibrium (HWE) test was performed to assess the genetic balance of the study population. The association between SNPs and IS risk was evaluated using binary logistic regression, with odds ratios (ORs) and 95% CIs. The analysis was adjusted for potential confounders, including age, sex, and diabetes mellitus.

## RESULTS

### Baseline data of the participants

The clinical characteristics of the case-control study participants are presented in Table 3. No significant differences were observed between the IS and control groups in terms of age, sex, height, weight, triglycerides, low-density lipoprotein, or apolipoprotein B levels. However, systolic blood pressure, diastolic blood pressure, and fasting blood glucose levels were significantly higher in the IS group than in the control group. In contrast, the control

**Table 1  Primer information for PCR amplification.**

| SNPs | Forward primer | Reverse primer |
|---|---|---|
| rs13005431 | 5′-GGAACCAGCAGG AGAAGAACA-3′ | 5′-GTAGATTAGTACC TTCAATGTC-3′ |
| rs4893819 | 5′-TTTGACACTCCCA GGATTTATG-3′ | 5′-TATTGCTCCCCTCC CTTTGA-3′ |
| rs6721961 | 5′-CCTTGCCCTGCTT TTATCTCA-3′ | 5′-CGCTTTGGTGGGAA GAGGT-3′ |
| rs35652124 | 5′-CCTTGCCCTGCTT TTATCTCA-3′ | 5′-CGCTTTGGTGGGAA GAGGT-3′ |
| rs6726395 | 5′-GTCAGTGTCAATC ATGCCAAG-3′ | 5′-GAGAGATACTTTTC ACGTGCC-3′ |
| rs2364723 | 5′-CTCTCCTAACCTTT CCTAACC-3′ | 5′-TTTCCTCTGTCCTGA CTGAAG-3′ |
| rs2706110 | 5′-CCCCTCAAAAACAG GAACTTG-3′ | 5′-GACCAATTCATACGA GGGAAC-3′ |

Notes.
PCR, polymerase chain reaction; SNP, single-nucleotide polymorphism.

**Table 2  Primer information for SNaPshot extension.**

| SNPs | Primer for extension | Extension direction |
|---|---|---|
| rs13005431 | 5′-TTTTTTTAGGGGGCCCACTGTTAAGGC-3′ | R |
| rs4893819 | 5′-ATTGTCTACCTTCTCTGATGTC-3′ | R |
| rs6721961 | 5′-TTTTTTTTTTTTTTTTTTTTTTTTTTTTT TTTTTTGCCTAGGGGAGATGTGGACAGC-3′ | F |
| rs35652124 | 5′-TTCGCAGTCACCCTGAACGCCC-3′ | F |
| rs6726395 | 5′-TTTTTTTTTTAATTATTCCATCCTACC CAAGC-3′ | F |
| rs2364723 | 5′-TTTTTTTTTTTTTTTTTTTTTTTTTTTTC CCAGGCTTGAGGAACAGTTAA-3′ | F |
| rs2706110 | 5′-TTTTTTTTTTTTTTTTTTATTAGTCATGG CATAGTTGAGA-3′ | F |

Notes.
F, forward; SNP, single-nucleotide polymorphism; R, reverse.

group exhibited higher total cholesterol, apolipoprotein A1, and high-density lipoprotein cholesterol levels.

## NRF2 SNPs and IS risk

Seven SNPs (rs13005431 (C/T), rs4893819 (C/T), rs6721961 (G/T), rs35652124 (T/C), rs6726395 (A/G), rs2364723 (C/G), and rs2706110 (C/T)) were in HWE and demonstrated good population representativeness. The genotype distribution and allele frequencies of these SNPs in the two groups are shown in Table S1. No significant differences were observed in the genotype and allele frequencies of *NRF2* rs13005431, rs4893819, rs6721961, rs6726395, rs2364723, and rs2706110 between the case and control groups (all $p > 0.05$). However, rs35652124 in *NRF2* was associated with IS susceptibility. In the co-dominant

**Table 3  Baseline data of study participants.**

| Characteristic | Control group (n = 141) | IS group (n = 159) | p-value |
|---|---|---|---|
| Age (years) | 60.000 [56.000, 68.000] | 63.000 [53.000, 71.000] | 0.922 |
| Sex (male/female) | 77/64 | 101/58 | 0.117 |
| Height (m) | $1.626 \pm 0.086$ | $1.644 \pm 0.085$ | 0.103 |
| Weight (kg) | $64.442 \pm 9.867$ | $64.182 \pm 12.546$ | 0.852 |
| SBP (mmHg) | $133.167 \pm 19.556$ | $140.365 \pm 23.208$ | **0.007** |
| DBP (mmHg) | $79.833 \pm 11.042$ | $84.522 \pm 14.382$ | **0.002** |
| FBG (mmol/L) | $5.863 \pm 1.929$ | $6.562 \pm 3.601$ | **0.035** |
| TCH (mmol/L) | $5.634 \pm 3.543$ | $4.571 \pm 1.383$ | **<0.001** |
| TG (mmol/L) | $2.089 \pm 2.262$ | $1.934 \pm 1.725$ | 0.507 |
| HDL-C (mmol/L) | $1.445 \pm 0.418$ | $1.150 \pm 0.835$ | **<0.001** |
| LDL-C (mmol/L) | $2.574 \pm 0.765$ | $2.411 \pm 0.927$ | 0.105 |
| Apo-A1 (g/L) | $1.274 \pm 0.238$ | $1.046 \pm 0.775$ | **0.001** |
| Apo-B (g/L) | $0.806 \pm 0.197$ | $0.797 \pm 0.278$ | 0.749 |

Notes.

Apo-A1, apolipoprotein A1; ApoB, apolipoprotein B; DBP, diastolic blood pressure; FBG, fasting blood glucose; HDL-C, high density lipoprotein cholesterol; IS, ischemic stroke; LDL-C, low-density lipoprotein cholesterol; TCH, total cholesterol; TG, triglyceride; SBP, systolic blood pressure.

Bold: $p < 0.05$.

**Table 4  Haplotype analysis of NRF2 polymorphism.**

| Haplotypes | Control group (N, %) | IS group (N, %) | Chi$^2$ | OR (95% CI) | p-value |
|---|---|---|---|---|---|
| CCTTGGC | 117 (41.6%) | 156 (49.1%) | 4.372 | 1.433 (1.022–2.008) | **0.037** |
| TCTTAGT | 44 (15.6%) | 43 (13.4%) | 0.473 | 0.851 (0.538–1.348) | 0.492 |
| CCTTGGC | 28 (10.0%) | 25 (7.7%) | 0.882 | 0.762 (0.431–1.346) | 0.348 |
| CTGTGGC | 22 (7.7%) | 18 (5.6%) | 2.422 | 0.722 (0.377–1.384) | 0.325 |
| TCTTAGC | 12 (4.4%) | 23 (7.3%) | 0.967 | 1.749 (0.858–3.563) | 0.120 |
| CTGCGGC | 14 (5.0%) | 15 (4.6%) | 0.037 | 0.930 (0.439–1.966) | 0.848 |
| CCTTGGC | 11 (4.0%) | 8 (2.5%) | 0.957 | 0.634 (0.253–1.591) | 0.328 |
| TTGCACC | 9 (3.3%) | 1 (0.3%) | 9.647 | 0.002 (0.000–0.030) | **0.002** |

Notes.

CI, confidence interval; IS, ischemic stroke; OR, odds ratio.

Bold: $p < 0.05$.

model, the TC genotype (TC vs. TT: OR = 1.869, 95% CI [.007–3.469], $p = 0.048$) increased the risk of IS.

## Haplotype analysis

Haplotype analysis identified eight haplotypes with distribution frequencies > 3% (Table 4). CCTTGGC was the most frequent haplotype, with a prevalence of 41.6% in the control group and 49.1% in the case group. This haplotype was found to be a risk factor for IS, increasing disease susceptibility by 43.3% ($p = 0.037$). Conversely, the TTGCACC haplotype may serve as a protective factor against IS.

**Table 5  Stratified analysis of rs35652124 in clinical features of patients with ischemic stroke.**

| rs35652124 | Classification | | OR (95% CI) | *p*-value |
|---|---|---|---|---|
| | N, % | N, % | | |
| Age (years) | <65 | ≥65 | | |
| TT | 15 (18.3%) | 13 (16.9%) | Ref | |
| TC | 35 (42.7%) | 43 (42.7%) | 1.418 (0.596–3.376) | 0.430 |
| CC | 32 (39.0%) | 21 (27.3%) | 0.757 (0.300–1.908) | 0.555 |
| Sex | Female | Male | | |
| TT | 9 (15.5%) | 19 (18.8%) | Ref | |
| TC | 29 (50.0%) | 49 (48.5%) | 0.800 (0.320–2.001) | 0.634 |
| CC | 20 (34.5%) | 33 (32.7%) | 0.782 (0.297–2.058) | 0.618 |
| Hypertension | No | Yes | | |
| TT | 13 (17.6%) | 15 (17.6%) | Ref | |
| TC | 39 (52.7%) | 39 (45.9%) | 0.867 (0.365–2.059) | 0.746 |
| CC | 22 (29.7%) | 31 (36.5%) | 1.221 (0.486–3.071) | 0.671 |
| Diabetes | No | Yes | | |
| TT | 20 (16.9%) | 8 (19.5%) | Ref | |
| TC | 60 (50.8%) | 18 (43.9%) | 0.750 (0.283–1.987) | 0.563 |
| CC | 38 (32.2%) | 15 (36.6%) | 0.987 (0.358–2.722) | 0.980 |

**Notes.**
CI, confidence interval; OR, odds ratio; Ref, reference.

## Stratified analysis of rs35652124 sites and clinical characteristics of patients with IS

Based on the latest edition of the China Stroke Prevention and Treatment Report 2020 (*Wang et al., 2022a*), we conducted a stratified analysis of potential confounding factors, including sex, age, history of hypertension, and diabetes mellitus, to assess their influence on the association between SNP sites and IS using a co-dominant model. No significant difference in the distribution frequency of the rs35652124 polymorphism were observed across stratified groups ($p > 0.05$). Thus, no interaction between the rs35652124 polymorphism and age, sex, hypertension, or diabetes mellitus in the co-dominant model (Table 5).

Dyslipidemia is a major risk factor for stroke. Thus, we investigated the association between rs35652124 and lipid levels in the peripheral blood of patients with IS. The patients with IS were divided into two groups: wild-type (TT genotype) and mutant-type (TC/CC genotype). Next, we compared the levels of various lipid indices between these two groups. As shown in Table 6, lipid levels did not significantly differ between the wild-type and mutant-type rs35652124 groups, indicating no correlation between rs35652124 and lipid levels ($p > 0.05$).

## Distribution analysis of rs35652124 across different regions and populations

Analyzing the distribution of rs35652124 across different regions and populations is valuable for understanding its influence on the incidence of IS in the Yunnan population. Globally, the frequencies of the rs35652124 T and C alleles are 62.44% and 37.56%,

**Table 6  Analysis of serum lipid levels in patients with rs65652124 and patients with ischemic stroke.**

| Indicator | rs35652124 | | t | p-value |
|---|---|---|---|---|
| | TT | TC/CC | | |
| TCH | 5.04 ± 1.91 | 4.47 ± 1.23 | 1.527 | 0.136 |
| TG | 2.78 ± 3.02 | 1.74 ± 1.24 | 1.778 | 0.086 |
| HDL-C | 1.08 ± 0.22 | 1.17 ± 0.92 | −0.517 | 0.606 |
| LDL-C | 2.64 ± 1.14 | 2.36 ± 0.87 | 1.473 | 0.143 |
| Apo-A1 | 0.99 ± 0.14 | 1.06 ± 0.86 | −0.413 | 0.680 |
| ApoB | 0.87 ± 0.28 | 0.78 ± 0.28 | 1.491 | 0.138 |

**Notes.**

Apo-A1, apolipoprotein A1; ApoB, apolipoprotein B; HDL-C, high-density lipoprotein cholesterol; LDL-C, low-density lipoprotein cholesterol; TCH, total cholesterol; TG, triglyceride.

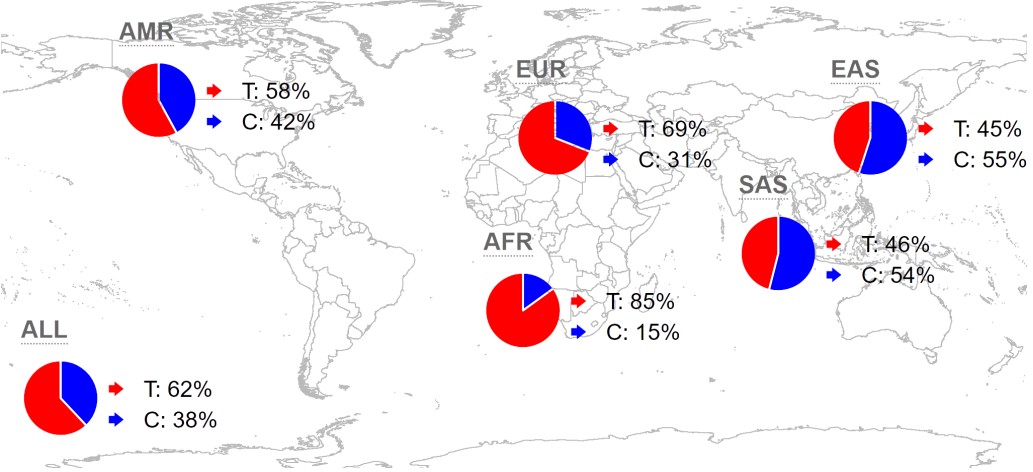

**Figure 1  Global distribution map of *NRF2* rs35652124 alleles.** AMR, American; EUR, European; AFR, African; SAS, South Asian; EAS, East Asian.

respectively. In the East Asian population, the T and C allele frequencies are 44.84% and 55.16%, respectively. The allele distribution in our study population aligns with the trends in East Asia. The distribution of this variant in other regions is shown in Fig. 1.

We also examined the distribution of the three rs35652124 genotypes across different populations. As shown in Fig. 2, genotype distribution varies significantly among populations. Table 7 indicates that the genotype distribution pattern in our study population is consistent with that of the Chinese Han population. However, significant differences were observed when compared with the Chinese Dai population in East Asia, the Finnish and British populations in Europe, the Colombian population in the Americas, and the Gambian and Yoruba populations in the Africa ($p < 0.05$).

## Analysis of NRF2 genetic variation and NRF2 levels in peripheral blood

As illustrated in Fig. 3A, NRF2 levels in the peripheral blood plasma of patients with IS (248.56 [191.63, 324.86] pg/mL) were significantly higher than those in the control group (173.34 [145.99, 214.82] pg/mL) ($p < 0.05$). To further investigate whether rs35652124

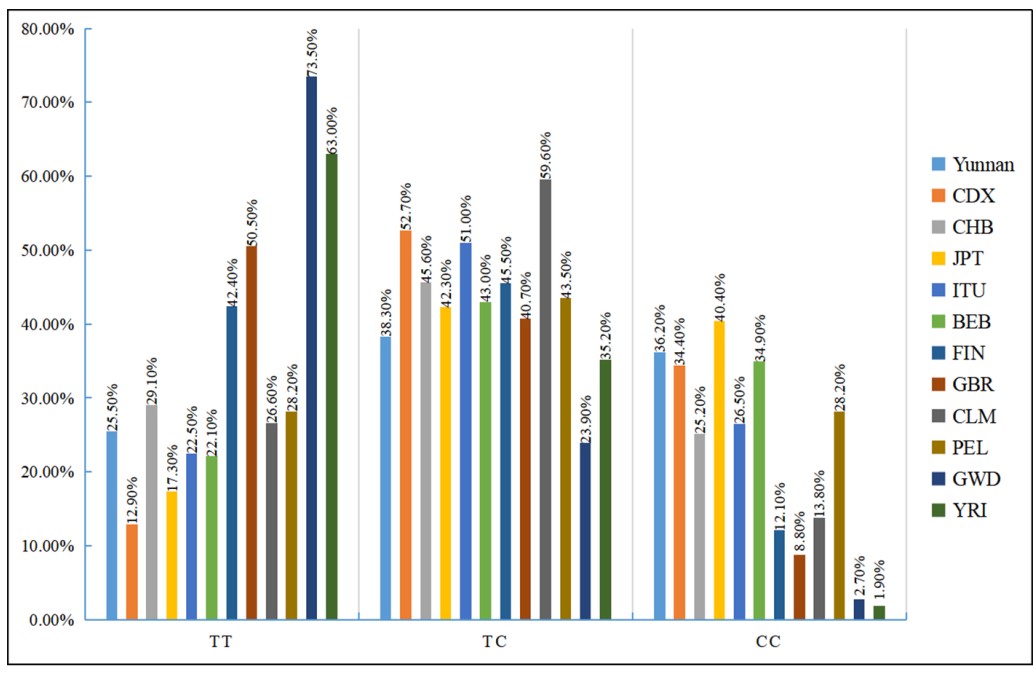

**Figure 2   Distribution histogram of three genotypes of NRF2 rs35652124 in different populations.**
BEB, Bengali in Bangladesh; CDX, Chinese Dai; CHB, Chinese Han in Beijing, China; CLM, Columbian; FIN, Finnish; GBR, British; GWD, Gambian; ITU, Indian Telugu in the UK; JPT, Japanese in Tokyo; PEL, Peruvian in Lima; Ref, reference;YRI, Yoruba.

**Table 7   Analysis of three genotypes of rs35652124 in different populations worldwide.**

| Region | Population | Frequency of genotyping (N, %) | | | $\chi^2$ | $p$-value |
|---|---|---|---|---|---|---|
| | | **TT** | **TC** | **CC** | | |
| East Asian | Yunnan | 36 (25.5%) | 54 (38.3%) | 51 (36.2%) | Ref | |
| | CDX | 12 (12.9%) | 49 (52.7%) | 32 (34.4%) | 7.042 | 0.030 |
| | CHB | 30 (29.1%) | 47 (45.6%) | 26 (25.2%) | 3.310 | 0.191 |
| | JPT | 18 (17.3%) | 44 (42.3%) | 42 (40.4%) | 2.357 | 0.308 |
| South Asian | ITU | 23 (22.5%) | 52 (51.0%) | 27 (26.5%) | 4.134 | 0.127 |
| | BEB | 19 (22.1%) | 37 (43.0%) | 30 (34.9%) | 0.583 | 0.747 |
| European | FIN | 42 (42.4%) | 45 (45.5%) | 12 (12.1%) | 18.644 | <0.001 |
| | GBR | 46 (50.5%) | 37 (40.7%) | 8 (8.8%) | 26.174 | <0.001 |
| American | CLM | 25 (26.6%) | 56 (59.6%) | 13 (13.8%) | 15.815 | <0.001 |
| | PEL | 24 (28.2%) | 37 (43.5%) | 24 (28.2%) | 1.513 | 0.469 |
| African | GWD | 83 (73.5%) | 27 (23.9%) | 3 (2.7%) | 67.969 | <0.001 |
| | YRI | 68 (63.0%) | 38 (35.2%) | 2 (1.9%) | 54.515 | <0.001 |

**Notes.**

BEB, Bengali in Bangladesh; CDX, Chinese Dai; CHB, Chinese Han in Beijing, China; CLM, Columbian; FIN, Finnish; GBR, British; GWD, Gambian; ITU, Indian Telugu in the UK; JPT, Japanese in Tokyo; PEL, Peruvian in Lima; Ref, reference; YRI, Yoruba.

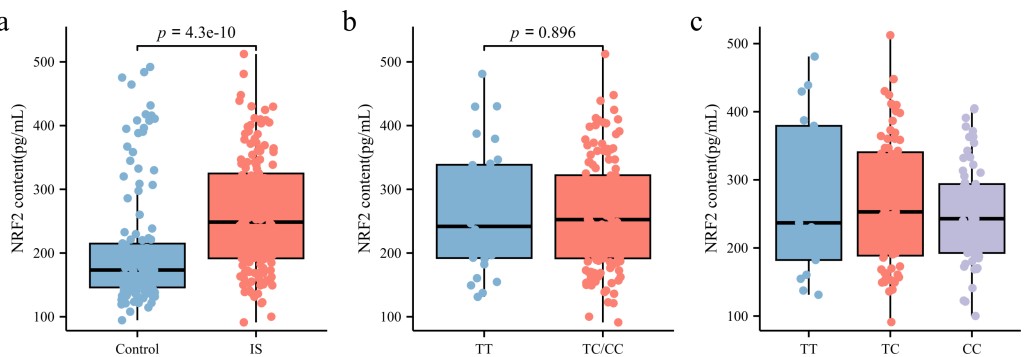

**Figure 3** **NRF2 content in the peripheral blood.** (A) Comparison of NRF2 content in the peripheral blood between the control and IS groups. (B) The content of NRF2 in peripheral blood of patients with ischemic stroke was compared according to the wild-type (TT genotype) and mutant-type (TC and CC genotype) of rs35652124. (C) The content of NRF2 in peripheral blood of patients with IS was compared according to the three genotypes (TT , TC and CC) of rs35652124.

polymorphisms influence NRF2 expression in patients with IS, we compared NRF2 levels among different genotypes at this locus. As shown in Figs. 3B–3C, neither homozygous nor heterozygous genotypes of rs35652124 significantly affected the NRF2 levels in the peripheral blood of patients with IS. These findings suggest that while genetic variation at rs35652124 is associated with IS susceptibility, it does not significantly modulate NRF2 expression levels in patients with IS.

## eQTL

An eQTL is a genetic variant that influences gene expression and plays a role in biological functions. We analyzed the tissue-specific expression of rs35652124 using the GTEx database. As shown in Fig. 4, rs35652124 affects NRF2 expression in the bladder, tibial nerve, thyroid, and several other tissues. However, no significant correlation was observed between rs35652124 and NRF2 expression in blood or brain tissues.

## Analysis of NRF2 genetic variation and ROS levels in peripheral blood

ROS levels in peripheral blood plasma was measured using ELISA. As shown in Fig. 5A, patients with IS had significantly higher ROS levels (160.3 [141.23, 194.61] pg/mL) than the control group (141.68 [100.43, 185.19] pg/mL) ($p < 0.05$). However, subgroup analysis based on rs35652124 genotype revealed no significant association between rs35652124 polymorphic variations and ROS expression levels (Figs. 5B–5C).

## DISCUSSION

Several studies have demonstrated that *NRF2* plays a crucial neuroprotective role in the development and progression of various diseases (*Liu, Locascio & Doré, 2019*). Endogenous activation of *NRF2* occurs following cerebral ischemia and helps prevent brain injury. Loss of *NRF2* in mice ($NRF2^{-/-}$) exacerbates the progression of IS-related brain injury, with *NRF2*-deficient mice exhibiting more severe cerebral infarctions on day 3 and poorer sensorimotor function on day 28 in a permanent ischemic model (*Jerotic et al., 2019*). The

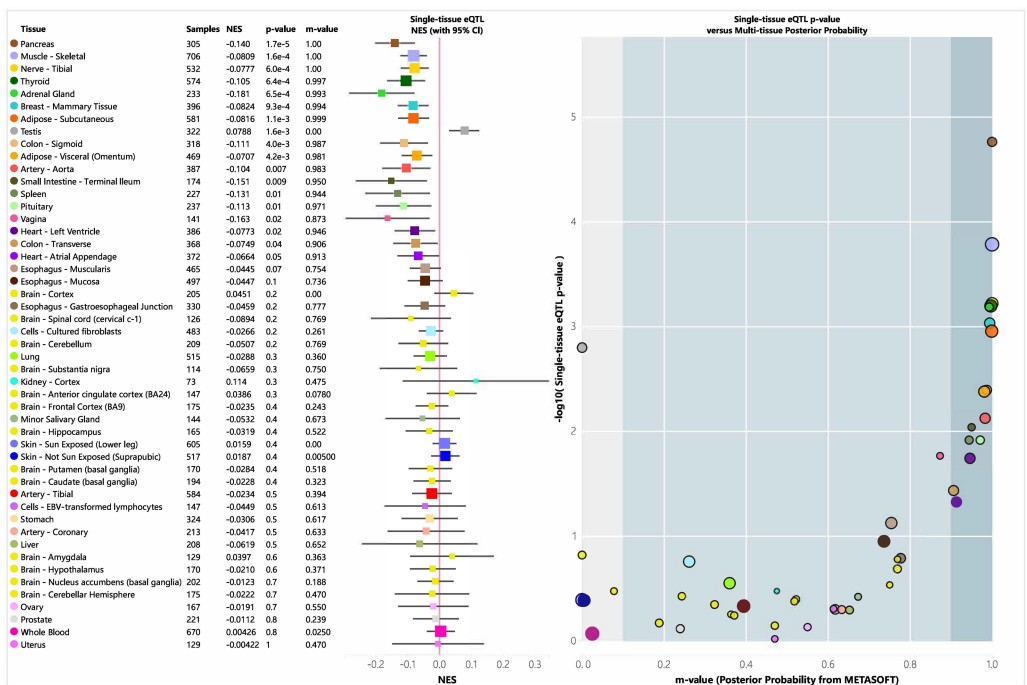

**Figure 4** **EQTL analysis of rs35652124.** CI, confidence interval; eQTL, expression quantitative trait locus; NES, Normalized Enrichment Score.

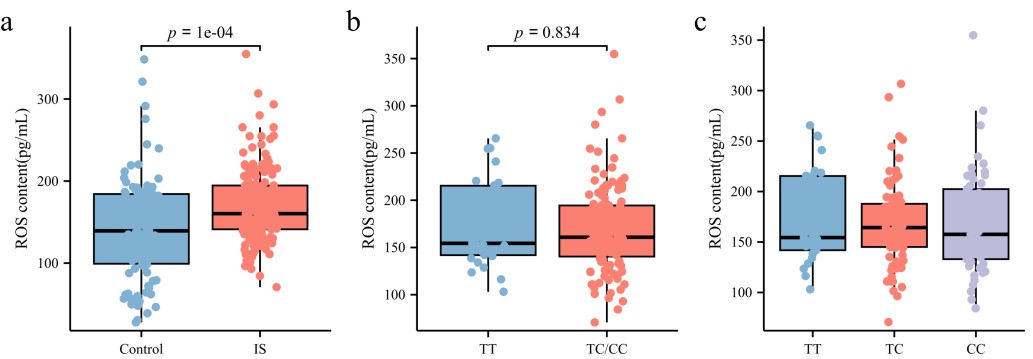

**Figure 5** **ROS content in the peripheral blood.** (A) Comparison of ROS content in the peripheral blood between the control and IS groups. (B) The content of ROS in peripheral blood of patients with ischemic stroke was compared according to the wild-type (TT genotype) and mutant-type (TC and CC genotype) of rs35652124. (C) The content of ROS in peripheral blood of patients with IS was compared according to the three genotypes (TT, TC and CC) of rs35652124.

absence of NRF2 reduces the body's and cells' antioxidant capacity and downregulates the downstream protective proteins, impairing the response to ischemia. In addition, aggravation of IS triggers inflammatory activation, further worsening ischemic injury and creating a vicious cycle. Previous research has indicated that polymorphisms in *NRF2* can influence its affinity, expression, and activity (*Song et al., 2016*). However, its relationship

with IS development among East Asian population has not been previously reported. This study revealed that *NRF2* polymorphism is associated with IS risk, with the TC genotype of rs35652124 in the co-dominant model potentially increasing IS susceptibility in a Chinese population from Yunnan.

Among conventional risk factors, hypertension is the most prevalent modifiable risk factor for IS, with up to 84% of patients with acute IS having hypertension (*McManus & Liebeskind, 2016*). Hypertension has also been identified as a key contributor to stroke-related mortality (*Pistoia et al., 2016*). Chronic hypertension remains a persistent long-term risk factor for acute IS. Elevated fasting blood glucose level indicate abnormal glucose metabolism and diabetes severity, increasing IS risk. Oxidative stress and mitochondrial dysfunction are critical mechanisms in hyperglycemia-induced ischemic brain injury. Under hyperglycemic conditions, excessive mitochondrial ROS production, impaired mitochondrial energy generation, and the release of apoptotic factors contribute to endothelial cell apoptosis and dysfunction, thereby exacerbating disease risk (*Zhang, Yan & Shi, 2013*). Furthermore, hyperlipidemia accelerates pathophysiological processes such as oxidative stress in endothelial cells, inflammation, and lipid peroxidation, all of which facilitate atherosclerosis development (*Alkahtani, 2022*). Hyperlipidemia also elevates IS risk by promoting conditions such as hypertension, insulin resistance, obesity, and cardiovascular and cerebrovascular diseases (*Kloska et al., 2020*). However, in this study, the observed variations in allele and genotype distribution of rs35652124 across different regions and populations may be attributed to genetic heterogeneity influenced by regional, environmental, ethnic, and lifestyle differences. Stratified analysis of rs35652124 revealed no significant association between its genotype and confounding factors such as age, sex, hypertension, diabetes, and lipid levels. These findings suggest that the association between rs35652124 and IS risk may not be significantly influenced by interactions with these factors.

Their pilot findings suggest that in children with ASD (autism spectrum disorder) the NEF2L2 rs35652124 polymorphism impacts adaptive responses that may potentially link to ASD severity, and genotype rs35652124 CC can be protective with respect to oxidative stress characteristic of the pathogenesis of autism (*Porokhovnik et al., 2023*). *Sorour et al. (2021)* reported that carriers of the T allele of rs35652124 were 3.329 times more likely to develop vitiligo than carriers of the C allele. In the PRIME study, rs35652124 was associated with aging-related phenotypes, including adverse drug reactions, clinical frailty scores, and multiple diseases (*Scutt et al., 2020*). Moreover, genetic variation at rs35652124 has been linked to the inflammatory characteristics of the liver and the development of cirrhosis in ALD (*Nunes dos Santos et al., 2019*). It is worth noting that our study found that the TC genotype in the co-dominant model may be a risk factor for IS, correlating with susceptibility to the disease. This heterozygous mutation has an impact on the occurrence of the disease, which may be caused by dose-dependent reversal. And the Epistasis caused by the possible interaction of homozygous mutations with variations of other genes makes the CC genotype not associated with the occurrence of IS.

Notably, a previous study using *in vitro* dual-luciferase reporter assay revealed that the rs35652124-C allele reduced NRF2 promoter activity and expression in human

microvascular endothelial cells (*Marczak et al., 2012*). However, the rs35652124 T > C variant is located in the core sequence of the putative binding site of transcription factor-specific protein 1 (SP1). The T > C change may enhance SP1 binding to the NRF2 promoter region, thereby increasing its activity and upregulating NRF2 expression. To investigate the mechanism by which *NRF2* rs35652124 influences susceptibility to IS, we examined whether this variant was associated with altered plasma NRF2 levels in patients with IS. Our results showed that NRF2 expression in peripheral blood was significantly higher in the IS group than in the control group. However, no significant differences in NRF2 content were observed among the three co-dominant genotypes of rs35652124. Further analysis using eQTL data confirmed that rs35652124 was not associated with plasma NRF2 levels. Notably, the potential effect of rs35652124 genotypes on intracellular NRF2 expression remains unverified due to the absence of cell-based experimental evidence in this study.

Previous studies have demonstrated that increased ROS levels in experimental IS models lead to endogenous *NRF2* activation, with significantly elevated NRF2 expression in ischemic infarction areas (*Farina et al., 2021*). We, therefore, analyzed the relationship between rs35652124 and ROS levels in peripheral blood. The results indicated no significant differences in ROS content among the three co-dominant genotypes of rs35652124. In IS, elevated ROS levels triggers *NRF2* activation. Since blood samples in this study were collected during the early stage of IS, any enhancement in antioxidant capacity due to *NRF2* activation may not have been evident. This finding suggests that a single genetic variation at rs35652124 does not directly alter ROS levels in the peripheral blood of patients with early IS. However, ROS content in the peripheral blood of the IS group was significantly higher than in the control group, likely due to cerebral ischemia-induced ROS accumulation, which activates endogenous oxidative stress and contributes to pathological damage.

Recent studies have also reported that specific *NRF2* haplotypes are associated with differences in promoter activity and the severity of chronic obstructive pulmonary disease (*Hua et al., 2010*). In our study, the CCTTGGC haplotype of *NRF2* (sequence: rs13005431–rs4893819–rs6721961–rs35652124–rs6726395–rs2364723–rs2706110) was found to increase IS risk.

Unfortunately, the other six genetic loci analyzed in this study did not show significant associations with IS occurrence. According to a study by *Nunes dos Santos et al. (2019)*, the rs4893819 locus in the *NRF2* promoter region is not associated with ALD susceptibility and has no effect on NRF2 expression or liver inflammatory activity. In addition, a genome-wide association study by *Maraganore et al. (2005)* found no link between rs13005431 and susceptibility to Parkinson's disease. In a study on ALD, rs6721961 had no specific correlation with ALD occurrence, and had no effect on liver inflammation in patients with ALD (*Nunes dos Santos et al., 2019*). The rs6726395 locus of *NRF2* has been linked to susceptibility to lung function impairment caused by smoking (*Masuko et al., 2011*). Genetic variants of rs2364723 are significantly associated with a reduced risk of CKD and may interact with rs35652124 to influence CKD risk (*Gómez-García et al., 2022*). The TAMRISK study found that the TT genotype of rs2706110 increased the risk of cerebrovascular diseases and suggested that insufficient NRF2 expression may contribute to their development (*Kunnas, Määttä & Nikkari, 2016*).

Thus, while *NRF2* polymorphisms have different effects on various diseases, further research is needed to determine their role in individual susceptibility to IS. A limitation of our study is that no functional analysis was conducted to elucidate the mechanism by which rs35652124 influences IS. Future studies should include dual-luciferase reporter assays to address this gap. Furthermore, the existence of dose–effect deficiency is explored through functional mechanism studies such as epigenetic regulation and protein interactions. Moreover, our study was limited to a single population. We plan to expand our research to multiregional and multiethnic populations to validate our findings and further explore genetic risk factors for IS.

## CONCLUSIONS

The polymorphic locus rs35652124 in *NRF2* was associated with susceptibility to IS in a Chinese population, with the TC genotype in the co-dominant model identified as a risk factor. However, the rs35652124 variant did not influence peripheral blood NRF2 levels or affect IS risk by altering ROS levels, suggesting that its role in IS susceptibility may not be mediated through oxidative stress. In addition, the CCTTGGC haplotype was identified as a potential risk factor for IS. While our findings highlight a genetic association between *NRF2* polymorphisms and IS, further functional studies are needed to elucidate the underlying mechanisms.

### Funding

This work was supported by the Natural Science Foundation of Yunnan Province (grant number 202001BA070001-133), Cultivating Plan Program for the Leader in Science and Technology of Yunnan Province (grant number D-2017057), Yunnan Natural Science Foundation of the Department of Education (grant number 2020J0590), and Science and Technology Project of Dali City (grant number 2021KBG083). The funders had no role in study design, data collection and analysis, decision to publish, or preparation of the manuscript.

### Grant Disclosures

The following grant information was disclosed by the authors:
The Natural Science Foundation of Yunnan Province: 202001BA070001-133.
Cultivating Plan Program for the Leader in Science and Technology of Yunnan Province: D-2017057.
Yunnan Natural Science Foundation of the Department of Education: 2020J0590.
Science and Technology Project of Dali City: 2021KBG083.

### Competing Interests

The authors declare there are no competing interests.

## Author Contributions

- Pengyu Wang conceived and designed the experiments, analyzed the data, authored or reviewed drafts of the article, and approved the final draft.
- Junxiu Lu performed the experiments, analyzed the data, prepared figures and/or tables, authored or reviewed drafts of the article, and approved the final draft.
- Min Wang performed the experiments, prepared figures and/or tables, and approved the final draft.
- Guangming Wang conceived and designed the experiments, authored or reviewed drafts of the article, and approved the final draft.
- Huaqiu Chen performed the experiments, prepared figures and/or tables, and approved the final draft.

## Human Ethics

The following information was supplied relating to ethical approvals (*i.e.*, approving body and any reference numbers):

This study was approved by the Medical Ethics Committee of The First Affiliated Hospital of Dali University (Ethical Application Ref: DFY20220415001).

## Data Availability

The raw data are available in the Supplementary Files.

## Supplemental Information

Supplemental information for this article can be found online at http://dx.doi.org/10.7717/peerj.19742#supplemental-information.

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
