# Peer review of "NRF2 polymorphism and susceptibility to ischemic stroke in a Chinese population"

_PeerJ, doi:10.7717/peerj.19742_

## Round 0.1 · original submission · Minor Revisions

Authors need to provide sample size calculations and inclusion/ exclusion criterion of study subjects selection. Additionally, post-hoc corrections should be applied for the multiple comparisons. Please address all comments given by the reviewers and submit point by point responses along with the revised manuscript.

·

Basic reporting

Several figures should be improved:
1. It is not clear from Fig 5,b and c, as well from Fig.3,b and c, whether the ROS and NRF2 content, respectively, in control vs ishemic stroke (IS) patients of alternative NRF2 genotypes are different ?
2.Fig 1 - "distributed alleles" (plural), not allele.

Experimental design

1.Lanes 255,266,327 posit that NRF2 expression in the of patients of different NRF2 genotypes with IS peripheral blood was not changed. However, in this paper only NRF expression in plasma (not in blood cells, as part of a blood) were studied. Potentially, differences between NRF2 genotypes might be exhibited in cells, however, this issue had not been studied in IS patients that should be mentioned as a limitation of the study design.
2. The regression analysis performed seem to belong to univariate one that should be stressed out.

Validity of the findings

It is not clear from the Table 4:
(a) whether the conclusion on CCTTGGC haplotype association with IS is valid after Bonferroni correction since univariate logistic regression has been used to compare the effect of each of eight haplotypes on IS frequency vs control;
(b) how haplotype frequencies were calculated (Chi square values are not shown).

Additional comments

In general, the paper contains novel data demonstrating for the first time a link between NRF2 genetic polymorphism and IS frequency. Most importantly, the results are indirectly confirmed by pathogenically related linkage of IS and NRF2 function as a master regulator of antioxidant responses in cells in pathological conditions related to IS severity. Absence of data on NRF2 expression in blood cells and pure discussion of possible mechanisms on relation of an NRF2 haplotype to protection from IS may limit the scientific soundness of the demonstrated novel data that may be improved by a more intensive discussion.

Reviewer 2 ·

Basic reporting

The manuscript presents an interesting and well-designed study examining the association between NRF2 polymorphisms and ischemic stroke (IS) risk in a Chinese population. The study employs sound methods for participant selection, genotyping, and statistical analysis, and offers meaningful insights into the genetic basis of IS susceptibility. The Materials and Methods section is comprehensive, but there are several areas where clarity, detail, and language could be improved to enhance the transparency and rigor of the work.

1.SNP Selection:
The SNP selection criteria are well laid out and the use of the dbSNP database is appropriate. However, it may be helpful to briefly explain why these particular SNPs were chosen. For instance, were they selected based on previous studies suggesting an association with ischemic stroke or NRF2 expression, or were they selected purely based on frequency and availability of data?
You might want to provide more information on the minor allele frequency (MAF) of these SNPs in the Chinese population specifically, as population-specific differences may influence the generalizability of your findings.
2.Participant Selection:
The inclusion criteria for the case group are adequately described, but more information on the selection process for control participants would strengthen the manuscript. Specifically, how were controls screened for other diseases or conditions that could confound the results? Clarifying these details would improve the transparency of participant selection.
It would also be useful to clarify how age and sex matching were conducted for the control group. If matching was based on specific criteria, this should be explicitly mentioned.
3.Statistical Analysis:
The statistical methods are mostly clear, but a few points require further clarification:
Logistic Regression: The use of logistic regression models to evaluate associations between SNPs and IS risk is appropriate. However, you should clarify whether you adjusted for any potential confounders in your regression models (e.g., age, sex, hypertension, diabetes, etc.). This can be stated as: “Logistic regression models were used to evaluate the associations between SNPs and IS risk, with adjustments for potential confounders such as age, sex, and comorbid conditions.”
4.language
For the methods and results sections of your paper, you should keep using past tense. It is recommended that you have your manuscript edited by a native English speaker.
5.Literature reference
Lines 134-135: ‘diagnostic criteria of the 2018 edition of the ìChinese Guidelines for the Diagnosis and
135 Treatment of Acute Ischemic Strokeî.’
Recommend citing relevant literature

Experimental design

no comment

Validity of the findings

no comment

---

## Round 0.2 · Major Revisions

Please address remaining comments of reviewer 1 and submit revised manuscript with pointwise responses.

·

Basic reporting

The authors significantly improved the text addressing the comments.

Experimental design

Authors improved the statistical treatment of the data as recommended, and provided an explanation for haplotypes.

Validity of the findings

The main finding that is intensively discussing in the paper is the establishing a link between ishemic stroke frequency and the NRF2 rs35652124 genotype. However, there is an invalid claim (lanes 211-212) that "In the co-dominant model, the TT genotype (TT vs. CC:
212 OR = 1.869, 95% CI = 1.007 3.469, p = 0.048) increased the risk of IS". Data in Supplemental Table 1 (copied from Suppl Table 1, below) do not confirm this statement

CC 51 53 Ref
rs35652124 Co-dominant CT 54 78 1.335 (0.704–2.532) 0.376
TT 36 28 1.869 (1.007–3.469) 0.048
Dominant CC 51 53 Ref

The claimed level of significant P (0.048) in reality belongs to another 2x2 Table pair, CT vs TT, not to TT vs CC (P value for the latter value is >0.4, according to GraphPad statistical software, www.graphpad.com/quickcalcs).Therefore, codominant model that is claimed by the authors cannot be considered showing that no association between ishemic stroke and any of NRF2 genetic variants studied have been found. Alternatively, a technical error in presentation of Table 1 Suppl data may occured.

Additional comments

I suggest to double check thoroughly the primary data to reveal where the technical error is, and present primary data for this part of the study for re-evaluation.
It is impossible to consider further the manuscript in its present form since the data on the main (and the only) finding are presented in an invalid manner.

---

## Round 0.3 · accepted · Accept

Your revised manuscript was reviewed by one of the original reviewers. Thank you for thoroughly addressing the reviewer's concerns, thus greatly improving your manuscript.

·

Basic reporting

No comment

Experimental design

No comment

Validity of the findings

No comment

Additional comments

The authors have significantly improved the paper, addressing the questions. Now it has become clear that according to the co-dominant model, the NRF2 TC genotype, as well as NRF2 haplotype, associates with IS development (TC vs. TT: OR = 1.869, 95% CI = 1.007-3.469, p = 0.048). Data demonstrating a link between ischemic stroke and NRF2 genetic polymorphism in the Chinese population are novel; conclusions are properly linked to thoroughly presented data.